# Global Perspectives on Patient Safety: The Central Role of Nursing Management

**DOI:** 10.3390/healthcare13243240

**Published:** 2025-12-10

**Authors:** Robert L. Anders

**Affiliations:** College of Nursing, University of Texas at El Paso, El Paso, TX 79968, USA; rlanders@utep.edu

**Keywords:** patient safety, nursing management, leadership, staffing, resilience, safety culture, WHO Global Patient Safety Action Plan, workforce sustainability, healthcare systems

## Abstract

**Highlights:**

**What are the main findings?**
Nursing management forms the structural foundation of patient safety systems worldwide.Transformational and authentic leadership enhance communication, teamwork, and psychological safety.Adequate staffing, supportive governance, and resilience programs reduce burnout and prevent harm.

**What are the implications of the main findings?**
Strong safety cultures emerge when leaders model transparency, just culture, and continuous improvement.Achieving WHO’s 2030 global safety vision depends on empowering nurse leaders at every level of health systems.

**Abstract:**

**Background:** Unsafe care remains a major global health challenge, contributing to millions of preventable deaths and ranking among the top ten causes of mortality and disability worldwide. The World Health Organization’s Global Patient Safety Action Plan 2021–2030 emphasizes the need for strong leadership and system-wide engagement to eliminate avoidable harm. As the largest component of the global healthcare workforce, nurses—especially those in management roles—are essential to achieving these goals. Objective: This narrative review synthesizes global evidence on how nursing management practices, particularly leadership, staffing, and safety culture, influence patient safety outcomes across diverse health systems. **Methods:** A purposive narrative review was conducted using PubMed, CINAHL, Scopus, and Web of Science databases. Peer-reviewed studies and organizational reports published between 2020 and 2025 were evaluated. A thematic synthesis approach was used to identify patterns related to leadership style, staffing ratios, workplace conditions, and organizational resilience. Quality appraisal followed adapted Critical Appraisal Skills Programme (CASP) and Joanna Briggs Institute (JBI) guidance. **Results:** A total of 37 peer-reviewed empirical studies were included in the narrative synthesis, along with key global policy and foundational framework documents used to contextualize findings. Evidence consistently demonstrated that transformational leadership, adequate nurse staffing, positive safety culture, and organizational learning structures are strongly associated with improved patient outcomes, reduced errors, and enhanced workforce well-being. Most studies exhibited moderate to high methodological rigor. **Conclusions:** Nursing management plays a decisive role in advancing global patient safety. Policies that strengthen leadership capacity, ensure safe staffing, promote just culture, and support nurse well-being are critical to achieving WHO’s 2030 safety objectives. Empowering nurse leaders across all regions is essential for building safer, more resilient health systems.

## 1. Introduction

Patient safety has emerged as one of the defining imperatives of twenty-first-century healthcare. The World Health Organization (WHO) estimates that unsafe care in hospitals causes 134 million adverse events and 2.6 million deaths annually in low- and middle-income countries (LMICs)—a scale of harm greater than that of malaria or tuberculosis. These events reflect the global reality that health systems designed to heal can also harm through failures in safety systems, staffing, and culture. Since the 1999–2000 Institute of Medicine report To Err Is Human, patient safety has shifted from an ethical obligation of individual clinicians to a systems-level responsibility of organizations and governments [1].

Among all professional groups, nurses occupy the most decisive position in determining whether safety practices translate from policy into daily behavior. As the largest sector of the global health workforce—representing nearly 59 percent of all providers—nurses coordinate, deliver, and evaluate care around the clock. Yet their effectiveness depends on nursing management, the mid-level leadership structure that translates strategy into action, aligns staffing with patient acuity, and models the communication behaviors that shape safety culture. The WHO Global Patient Safety Action Plan 2021–2030 explicitly identifies the nursing workforce and its leadership as essential to achieving its vision of eliminating avoidable harm in healthcare. Earlier systematic reviews (2015–2019) linked leadership to safety culture but did not address post-pandemic resilience or digital health innovations. This review updates and extends those findings by synthesizing evidence generated from 2020 to 2025 [2,3,4].

### 1.1. Conceptual Foundations

Two classic frameworks illuminate how nursing management contributes to safer systems. First, Donabedian’s structure–process–outcome model (1966) conceptualizes quality of care as the interaction of organizational structures (e.g., staffing ratios, leadership hierarchy), care processes (communication, supervision, adherence to protocols), and outcomes (infection rates, mortality, satisfaction) [5]. Nurse managers operate at the nexus of these three domains: they design the structures (policies, staffing grids), monitor the processes (rounds, huddles, incident review), and interpret outcomes for improvement.

Second, Reason’s “Swiss-Cheese” model of system failure (1990) provides a complementary lens [6]. It posits that errors occur when multiple layers of defense—policies, technology, supervision—develop latent weaknesses that align to permit harm. Effective nursing management adds additional layers of resilience through supervision, open communication, and rapid problem-solving. When leaders encourage reporting and learning rather than blame, they reduce the likelihood that latent failures will align into catastrophic events.

Together, these theories underscore that safety is not primarily a matter of individual nurses’ vigilance but of organizational design and leadership. Donabedian emphasizes alignment of structure and process; Reason emphasizes detection and recovery from error. Both depend on competent, empowered nurse leaders.

### 1.2. Global Context and Persistent Disparities

While high-income countries (HICs) have introduced robust safety infrastructures—accreditation standards, electronic incident reporting, and continuous-quality programs—LMICs continue to face profound resource and workforce shortages [4,7]. In many African and South-Asian settings, nurse-to-patient ratios exceed 1:15, documentation is paper-based, and punitive responses to error discourage transparency across many settings, including hierarchical hospital cultures where nurses report significant second-victim distress and hesitancy to disclose incidents [8]. Safety-culture surveys in both primary care and hospital systems of the Middle East illustrate how variation in leadership support and reporting climate can widen disparities in preventable harm [9,10]. Yet even within resource-limited environments, strong local leadership has demonstrated measurable safety gains. In Ghana, for example, structured nurse-led safety huddles reduced medication error rates by 22 percent within 6 months [11]. Similarly, in Thailand, leadership development workshops emphasizing psychological safety improved incident reporting by 38 percent [2]. These examples affirm that culture and leadership can mitigate structural deficits.

The COVID-19 pandemic further magnified the connection between safety, staffing, and workforce well-being. Studies across multiple countries revealed surges in nurse burnout, moral distress, and turnover intentions, directly linked to perceived managerial support [12,13,14,15,16]. Evidence from U.S. hospitals also shows that pre-pandemic nursing practice environments strongly predicted both nurse safety and adverse events during later crisis conditions, underscoring the long arc of managerial responsibility [17,18]. Hospitals where nurse managers maintained visible communication, equitable scheduling, and access to mental health resources experienced lower infection rates and higher retention. The pandemic thus reinforced an urgent global lesson: patient safety is inseparable from staff safety [13,14,16,19].

### 1.3. Rationale and Objectives

Although hundreds of studies address safety outcomes, few synthesize the management mechanisms that enable success across settings. The purpose of this narrative review is therefore to analyze global evidence (2020–2025) linking nursing management practices—leadership, staffing, and resilience—to patient-safety outcomes, and to interpret these relationships through established theoretical and policy frameworks. The review situates nursing management as the bridge between global policy aspirations and local bedside realities, demonstrating that without effective leadership, no safety intervention can be sustained.

## 2. Methods

### 2.1. Design

A narrative review methodology was selected to integrate empirical findings and conceptual perspectives across heterogeneous contexts. Unlike a systematic review that seeks exhaustive coverage, a narrative synthesis allows exploration of patterns, theoretical linkages, and contextual nuances—essential for a global topic encompassing diverse health systems. The review followed the Joanna Briggs Institute’s guidance for narrative reviews. It aligned its analytical focus with the WHO Global Patient Safety Action Plan 2021–2030s strategic objectives of leadership, workforce engagement, and learning culture [2,4,20].

This review followed the Joanna Briggs Institute (JBI) narrative review framework, which emphasizes transparency in the identification, appraisal, and synthesis of evidence.

### 2.2. Review Question

PICo elements: Population—nurses and nurse managers; Phenomenon of Interest—management practices influencing patient safety; Context—global healthcare systems. Review question: How do nursing-management practices affect patient-safety outcomes across diverse contexts?

### 2.3. Data Sources and Search Strategy

Electronic searches were conducted in PubMed, CINAHL, Scopus, and Web of Science for studies published in English between January 2020 and March 2025. Search strings combined controlled vocabulary (MeSH/CINAHL Headings) and keywords related to “nursing management” OR “nurse leadership” AND “patient safety”, “staffing ratios” OR “missed care” OR “safety culture”; and “WHO Global Patient Safety Action Plan”. Boolean operators and truncation were applied to capture variants. Gray literature from the WHO, the OECD, and national health system reports were reviewed to contextualize empirical results [4,20]. The three-stage search approach recommended by JBI was applied: (1) an initial limited search to identify keywords and index terms; (2) a comprehensive search across all databases using the finalized strategy; and (3) screening of reference lists for additional sources. An example of the complete PubMed search string is provided in Appendix A. Reference lists of retrieved papers were also hand-searched. The complete database search strategy and PRISMA flow diagram are provided in Appendix A.

The search process yielded 287 articles across all databases. After removing 63 duplicates, 224 unique titles and abstracts were screened for relevance. After applying the inclusion and exclusion criteria described below, 68 articles underwent full-text review. One article was excluded due to an inaccessible full text, resulting in 37 peer-reviewed empirical studies that met all requirements and informed the thematic synthesis. Eight authoritative organizational reports and policy/framework documents (WHO, OECD, ICN, and classic safety/quality models) were purposively included to contextualize findings [4,20]. While all sources informed the analysis, the final manuscript cites the most relevant evidence supporting the three emergent themes—leadership and safety culture, staffing and patient outcomes, and workforce resilience and technology integration—yielding a reference list that includes the 37 empirical studies plus selected policy and foundational sources.

### 2.4. Inclusion and Exclusion Criteria

Eligibility criteria were defined according to the PICo framework for narrative reviews. The population included registered nurses, nurse managers, and nursing leaders at all organizational levels. The phenomenon of interest encompassed nursing management practices—including leadership styles, staffing models, safety culture initiatives, workforce resilience programs, and technology integration—that were explicitly linked to patient safety outcomes such as adverse events, mortality rates, safety culture scores, and missed nursing care. The context included all healthcare settings (acute care, long-term care, community health) across all geographic regions and healthcare system types.

Additional publication characteristics required that studies: (1) employed quantitative, qualitative, or mixed-methods designs, or were systematic/narrative reviews with empirical synthesis; (2) were published in peer-reviewed English-language journals between January 2020 and March 2025; and (3) reported original data or authoritative policy evidence. Excluded were commentaries without data, studies focusing exclusively on medical leadership, and quality improvement initiatives without explicit patient safety metrics.

### 2.5. Data Extraction and Quality Appraisal

Quality assessment followed adapted criteria from the Critical Appraisal Skills Programme (CASP) checklists for observational and qualitative studies. The CASP tool was selected due to its flexibility in accommodating heterogeneous study designs. Eighty-two percent of included studies met moderate-to-high rigor criteria with clear sampling and analytic coherence [2].

Each article was screened independently by the author using titles, abstracts, and full-text reviews. Key variables extracted included study design, country, sample size, level of analysis, leadership variables, staffing metrics, safety outcomes, and principal findings.

### 2.6. Data Synthesis and Thematic Coding

Given the diversity of designs and outcome measures, data were coded inductively using Thomas and Harden’s thematic synthesis approach. Three dominant themes emerged: leadership and safety culture; staffing adequacy and workload; and workforce resilience and technology integration. These were mapped onto the Donabedian model (structure to process to outcome) [5].

### 2.7. Trustworthiness and Rigor

Analytical transparency was maintained through an audit trail documenting search strings, inclusion decisions, and coding revisions. Findings were compared with WHO safety and workforce guidance, OECD Economics of Patient Safety reports, and ICN workforce briefs [4,20].

### 2.8. Ethical Considerations

Because this review synthesized publicly available literature, institutional review board approval was not required.

## 3. Results

### 3.1. Leadership and Safety Culture

Leadership emerged as the most decisive determinant of patient-safety culture across all regions. Cross-sectional studies demonstrated that when nurse managers modeled safety-focused behaviors—open communication and non-punitive responses to errors—unit-level safety metrics improved [2,3,21,22,23]. Transformational leadership was repeatedly associated with higher safety-culture scores, improved communication, and lower event rates; one study found a one-point increase in transformational leadership correlated with a 0.26-point increase in safety culture (*p* < 0.01) [22,23].

Leadership functions as both structure and process within Donabedian’s and Reason’s models, thickening organizational defenses by fostering trust and reporting [5,6]. Conversely, authoritarian styles permit latent failures to align [6].

Authentic leadership reduced turnover intentions among novice nurses, indirectly strengthening safety through workforce stability [24]. Recent multi-country work also links authentic nurse leadership with stronger safety climates and fewer reported errors [25]. Servant-leadership interventions emphasizing humility and listening improved near-miss reporting [2].

Cross-cultural evidence shows leadership is adaptable. Nordic shared-leadership councils reduced falls, whereas hierarchical systems often inhibit voice. Targeted leadership programmes emphasizing respect have improved safety climates, with national nurse-leadership curricula producing measurable gains [2,26].

During COVID-19, visible and compassionate leadership buffered burnout and safety lapses; nurses rating managerial communication as excellent were less likely to report burnout and documentation errors [13,14].

### 3.2. Staffing and Patient Outcomes

Adequate nurse staffing remains the most evidence-based predictor of patient safety. Each additional patient per nurse increases mortality and failure-to-rescue, while safer ratios reduce infections and medication errors [27,28,29,30].

When staffing structures collapse, care processes truncate, producing missed care that mediates mortality risk [31,32,33].

HICs advancing minimum ratios report outcome gains, while LMICs face shortages and migration pressures. The global deficit remains concentrated in sub-Saharan Africa and Southeast Asia [7]. Skill mix matters; higher RN proportions lower mortality [29].

Technology can mitigate workload via acuity-based rostering and staffing alerts but requires managerial interpretation [34,35].

### 3.3. Workforce Resilience and Technology Integration

Resilience has become a critical safety pillar. Burnout affects roughly a third of nurses worldwide and is linked to poorer safety climates and more adverse events; resilience rebuilds attention, empathy, and decision-making through psychological resources and organizational support [13,16].

#### 3.3.1. Individual-Level Strategies

Mindfulness training, reflective journaling, and peer-support groups reduce emotional exhaustion by 20–30 percent [14,16,19]. Emotional-intelligence programmes strengthen situational awareness and can improve early detection of risk, but individual resilience strategies cannot compensate for unsafe systems [36].

#### 3.3.2. Organizational-Level Strategies

Leadership walk-rounds that listen to staff concerns act as diagnostic tools for system stress. Institutions implementing structured well-being rounds reported fewer medication errors and lower absenteeism; flexible scheduling, rest breaks, and mental-health access further buffer fatigue [14,16,19]. Egypt-based leadership–resilience evidence and Chilean nursing safety-culture findings link resilience-oriented leadership to safety outcomes [37,38]. Communication breakdowns are consistently identified as major contributors to preventable harm; in critical care settings, poor teamwork and weak reporting climates are associated with higher adverse-event rates [3,39]. Nurse managers who facilitated daily briefings and real-time feedback loops achieved greater adherence to safety protocols [2,3].

#### 3.3.3. Technological Integration

Digital systems enhance safety when human centered. EHR decision supports reduce transcription errors but risk alert fatigue unless nurses are involved in design [34]. Digital health technology interventions for medication safety, including electronic reconciliation, smart dispensing, and decision-support platforms, show growing clinical and economic value, though evidence quality varies across settings [35]. Prospective hazard-analysis methods such as FMEA increasingly support nurse-manager identification of system vulnerabilities before harm occurs [40]. Electronic medication reconciliation reduces medication discrepancies and adverse drug events in U.S. hospitals [41]. In Brazil, Tele-ICU platforms enable remote intensivists to guide bedside teams through daily tele-rounds and feedback [42].

#### 3.3.4. Post-COVID Global Recovery

The pandemic catalyzed collaboration and renewed emphasis on psychological safety. Health systems increasingly track workforce well-being alongside patient-safety indicators, and nurse managers lead debriefs after crises to convert stress into organizational learning [3,4,16].

In sum, workforce resilience and technology integration represent critical frontiers of nursing-management responsibility. A resilient, digitally supported workforce is not ancillary to safety; it is the mechanism through which safety is continuously produced and renewed.

## 4. Discussion

Patient safety represents both a scientific discipline and a moral contract between healthcare professionals and the populations they serve. The findings of this review reaffirm that nursing management—through leadership, staffing oversight, and workforce resilience—remains the critical mechanism for translating safety principles into practice. These themes converge in the global policy arena, where WHO, OECD, and ICN all call for strategic empowerment of nurse leaders as architects of safe systems [4,20].

### 4.1. Linking Evidence to Global Policy Frameworks

The WHO Global Patient Safety Action Plan 2021–2030 outlines strategic objectives that depend directly on nursing management [4]. Building high-reliability organizations and creating cultures of safety require leaders who operationalize these aims at the unit level. The OECD’s Economics of Patient Safety further quantifies the cost of unsafe care, estimating that preventable harm consumes substantial hospital expenditures [20]. From a stewardship perspective, investment in nurse-management training and staffing infrastructure yields high value by preventing costly adverse events [20,27,29].

### 4.2. Ethical and Humanistic Dimensions

Patient safety is grounded in nonmaleficence, justice, and fidelity. Nurse leaders embody these values by creating systems that prevent suffering and promote equity. Workforce inequity constitutes an ethical fault line in global safety. Understaffing, moral distress, and burnout disproportionately affect nurses in under-resourced regions and marginalized populations within wealthier systems. Protecting clinician well-being is a patient-safety and ethical imperative equal to protecting patients; when nurses work under conditions that compromise their own safety, the moral contract of care is violated [36].

### 4.3. Implementation Challenges and Sustainability

Implementing patient-safety reforms demands not only technical knowledge but also organizational change. Resistance to reporting systems, limited budgets, and policy discontinuity can derail initiatives. Donabedian’s model implies that structural improvements (such as staffing laws) must be paired with process change (communication, feedback, and learning) to yield durable outcomes [5]. Sustainability hinges on continuous education and leadership continuity. Nursing organizations and policy-advocacy coalitions help sustain staffing and safety reforms across political cycles [4,29,43].

### 4.4. Technological and Post-Pandemic Evolution

Digital transformation offers opportunities and risks. Artificial intelligence–driven decision support and automated medication systems can reduce error probability but introduce new failure modes. Nurse managers must mediate between technology and practice, ensuring usability, training, and data integrity. Human-centered design minimizes alert fatigue and cognitive overload [34,35].

### 4.5. Practical Applications for Nurse Managers

Evidence across settings yields several actionable strategies embedded in nursing management (Table 1).

Evidence levels reflect qualitative judgment based on methodological rigor and study design; no formal GRADE framework was applied (Figure 1 and Table 2).

### 4.6. Toward a Global Culture of Safety

The cumulative evidence suggests that the future of patient safety lies in interdependence—between nations, professions, and human–technology systems. Nursing management forms the connective tissue of this interdependence. When nurse leaders are empowered to allocate resources, shape policy, and nurture resilience, safety evolves from compliance to culture. As the WHO Action Plan envisions, every health facility should operate as a learning organization that detects, prevents, and learns from error. Nurse managers are custodians of this learning [4].

Figure 2 situates nursing management at the center of patient-safety systems, integrating structural, process, and outcome dimensions based on Donabedian’s model and Reason’s Swiss-Cheese theory. Three interrelated domains—staffing and resource adequacy, leadership and safety culture, and workforce resilience with technology integration—form the core management practices that determine safety outcomes.

Domain 1: Staffing and Resource Adequacy represents the structural foundation. Adequate nurse-to-patient ratios, appropriate skill mix, and workload balance enable effective safety processes. Insufficient resources create system vulnerabilities, increasing the likelihood of adverse events.

Domain 2: Leadership and Safety Culture embodies the process dimension, where transformational and ethical leadership behaviors promote psychological safety, communication, and continuous learning. Nurse managers operationalize policies, model transparency, and reinforce accountability, closing gaps that could align into system failures.

Domain 3: Workforce Resilience and Technology Integration functions as the adaptive layer that sustains safety under dynamic pressures. Resilience programs, emotional-intelligence development, and human-centered digital tools create flexible defenses that detect and mitigate risk in real time.

These domains interact within organizational and policy environments influenced by national regulation, accreditation standards, and global frameworks such as the WHO Global Patient Safety Action Plan 2021–2030. Arrows flow outward from nursing management to depict causal pathways from structure and process to improved outcomes—reduced adverse events, lower mortality, enhanced patient satisfaction, and increased workforce stability. Feedback loops signify continuous learning, emphasizing safety as an evolving system property maintained through leadership, reflection, and innovation.

## 5. Limitations

This review was conducted by a single reviewer and restricted to English-language publications, which may have limited the inclusion of regional perspectives. Nevertheless, all steps were transparently documented in accordance with JBI guidance, and appraised studies met minimum quality standards using CASP. The reliance on a narrative-review approach may introduce interpretive bias; however, this design allowed integration of diverse empirical and policy sources. Future research should employ collaborative multi-reviewer processes, include non-English databases, and apply standardized safety-culture metrics for cross-national comparison.

## 6. Conclusions

Unsafe care continues to represent one of the most significant and preventable threats to health worldwide. Yet the cumulative evidence underscores that many of these harms are avoidable when nurse managers are empowered to lead. Through adequate staffing, supportive leadership, and a culture of safety, nurse managers act as the vital bridge between policy and practice. Across nations and healthcare systems, patterns remain consistent: transformational and ethical leadership cultivates trust and transparency; optimal staffing levels safeguard both patients and nurses; and workforce resilience sustains performance under strain. When health systems align these principles with human-centered technology and continuous learning, they achieve safer outcomes and stronger workforce stability. To realize the WHO’s vision of eliminating avoidable harm by 2030, governments, educators, and healthcare organizations must recognize that nurse leaders are architects of safety. Investing in nursing management means investing in the human relationships that anchor healing itself.

## Figures and Tables

**Figure 1 healthcare-13-03240-f001:**
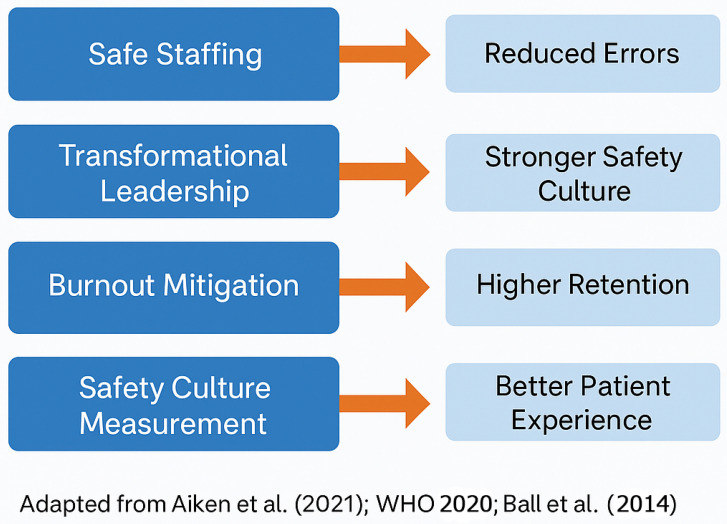
Evidenced Based Nursing Management Strategies for Safer Care. Adapted from [4,27,31].

**Figure 2 healthcare-13-03240-f002:**
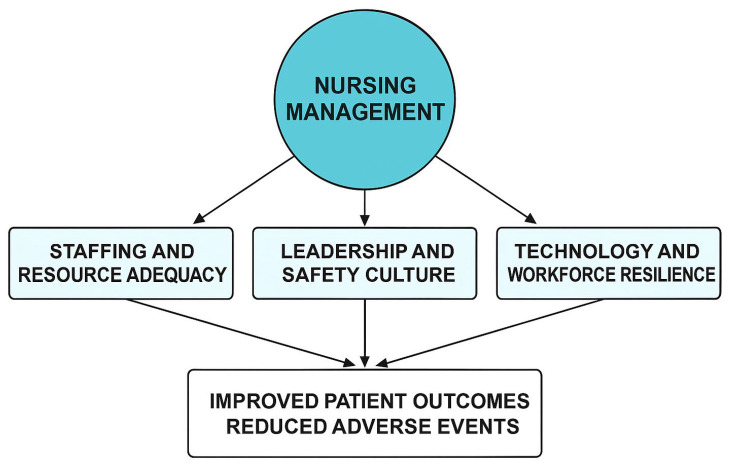
Conceptual Framework of Nursing Management and Patient Safety (description).

**Table 1 healthcare-13-03240-t001:** Evidence-Based Recommendations for Nurse Managers.

Management Strategy	Evidence Level	Practical Implementation Steps
Safe staffing ratios	High (Aiken et al., 2021; Griffiths et al., 2019; McHugh et al., 2021) [27,28,29]	Maintain ≤ 4 patients per RN on med–surg units; monitor overtime < 10%.
Transformational-leadership training	Moderate (Boamah et al., 2018; Hamdan et al., 2024; Bernardes et al., 2025) [22,23,25]	Integrate leadership modules, coaching, and peer-mentoring.
Safety-culture measurement	High (Alfadhalah et al., 2021; Abuosi et al., 2022; Hesgrove et al., 2024) [3,9,11]	Conduct regular HSOPS-based or equivalent safety-culture surveys and debrief unit-level action plans.
Missed-care reduction programmes	High (Ball et al., 2014; Bagnasco et al., 2020; Senek et al., 2020) [31,32,33]	Apply MISSCARE or similar tools; track missed-care items and adjust staffing and workflows.
Burnout and fatigue mitigation	Moderate (Welp et al., 2023; Galanis et al., 2021; Havaei et al., 2021; White et al., 2019; Li et al., 2024) [13,14,15,16,19]	Implement resilience rounds and mental-health supports; monitor burnout and turnover intention.
Technology safety integration	Moderate (Carayon et al., 2014; Insani et al., 2025; Cui et al., 2025; Tamblyn et al., 2019; Pereira et al., 2024) [34,35,40,41,42]	Involve nurses in EHR design; monitor alert fatigue; evaluate digital safety tools, medication reconciliation, and tele-ICU support.
Continuous learning systems	Moderate (Kohn et al., 2000; OECD 2022; WHO 2020) [1,4,20]	Establish daily huddles and unit-level safety debriefings; integrate learnings into ongoing improvement cycles.

**Table 2 healthcare-13-03240-t002:** Global Policy and Practice Implications.

Domain	Global Action Required	Potential Outcomes
Leadership development	National leadership academies; integration of evidence-based leadership frameworks; inclusion of nurses on national and international policy boards [2,4]	Stronger safety culture; reduced regional and institutional variance; improved strategic decision-making.
Workforce sustainability	Implementation of safe-staffing legislation; global alignment of nurse-migration policies; investment in retention and mental-health supports [4,7,27,29]	Lower turnover; improved retention in LMICs; enhanced workforce stability and patient outcomes.
Ethics and governance	Embedding just-culture principles into accreditation and regulatory standards; strengthening organizational accountability structures [1,36]	Greater transparency; reduction in punitive climates; increased reporting of safety incidents.
Technology and data	Investment in human-centered digital systems; reducing alert fatigue; expanding interoperability and real-time data analytics [34,35,40]	Real-time monitoring; predictive risk analytics; improved medication and documentation safety.
Education and research	Standardized global metrics linking leadership, well-being, and safety; expansion of cross-national research consortia [2,4,20]	Robust benchmarking; stronger policy evidence base; accelerated adoption of best practices.

## Data Availability

No new data were created or analyzed in this study. Data sharing is not applicable to this article.

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
