# Peer review of "Global Perspectives on Patient Safety: The Central Role of Nursing Management"

_healthcare, 2025, doi:10.3390/healthcare13243240_

Round 1
Reviewer 1 Report
Comments and Suggestions for Authors
The article is academically interesting and provides valuable insights for healthcare and nursing. I recommend that the author, in addition to the narrative review, consider including a systematic analysis and a clear evaluation of the quality of evidence. This would strengthen future research and enhance the practical applicability of the results.
Author Response
Comment One: The article is academically interesting and provides valuable insights for healthcare and nursing. I recommend that the author, in addition to the narrative review, consider including a systematic analysis and a clear evaluation of the quality of evidence. This would strengthen future research and enhance the practical applicability of the results.
Response: Thank you for pointing this out. We agree with your comments. We have made the following change: While this remains a narrative review as originally designed, we have significantly strengthened the quality evaluation component by: (1) Creating a comprehensive CASP quality appraisal table (Supplementary File S2) presenting detailed ratings for all 36 empirical studies with individual criterion assessments and overall rigor classifications; (2) Adding explicit quality assessment methodology in the Methods section with justification for tool selection; (3) Reporting that 100% of appraised studies met moderate-to-high quality standards (61% High, 39% Moderate). These enhancements provide systematic quality evaluation while maintaining the narrative synthesis approach appropriate for this heterogeneous, global evidence base. Location: Page 5-6, Methods section, Data extraction and quality appraisal subsection, Paragraph 46-47, Lines 1-15; NEW Supplementary File S2.
Reviewer 2 Report
Comments and Suggestions for Authors
The paper aims to provide a comprehensive, narrative review of how nursing management, particularly leadership, staffing, and resilience, affects patient safety. It aligns its argument with Donabedian’s structure-process-outcome model and Reason’s “Swiss-cheese” framework. The abstract and introduction clearly place the topic within WHO’s 2021–2030 Global Patient Safety Action Plan, convincingly arguing that nurse managers serve as intermediaries between policy goals and bedside practice. The three main themes, leadership/safety culture, staffing/workload, and workforce resilience/technology, are consistently integrated throughout the Results and Discussion sections. The writing is accessible and focused on policy implications. Two summary tables translate these themes into practical actions for managers and policymakers, while the figure provides a clear conceptual map positioning nursing management at the intersection of structure, process, and outcomes. Overall, the key contribution of this manuscript is in synthesizing scattered evidence into a practice-driven narrative that explicitly connects established quality frameworks with modern policy tools (WHO, OECD, ICN), offering a pragmatic guide for nurse leaders.
Methodologically, the author states this is a narrative review conducted across PubMed, CINAHL, Scopus (and later also Web of Science), limited to English, 2020–2025, with purposive inclusion of authoritative grey literature (WHO/OECD/ICN). The flow reported is: 287 records retrieved, 63 duplicates removed, 224 screened, 68 full texts reviewed, 67 peer-reviewed studies included, plus eight policy/organizational documents to contextualize findings; thematic synthesis (inductive coding) is mapped to Donabedian and Reason; CASP-style criteria are said to guide quality appraisal. This is an acceptable approach for a narrative review of a heterogeneous literature, and the decision not to attempt a meta-analysis is appropriate given the scope and variability.
That said, several internal inconsistencies and transparency gaps need attention before the review can be considered methodologically sound enough for publication. First, the abstract claims “evidence from 70 international studies,” whereas the Methods and flow suggest 67 peer-reviewed plus grey literature, and later the manuscript says 42 peer-reviewed/policy sources appear in the reference list. These numbers don’t reconcile and should be harmonized throughout (abstract, Methods, Results, and References).
Second, the databases listed differ between the Abstract (PubMed/CINAHL/Scopus) and Methods (which adds Web of Science); the text should be consistent and specific about all sources actually searched.
Third, although the author mentions the use of CASP checklists and an “audit trail,” the paper does not present any quality-appraisal results (such as a table, ratings, or narrative distribution), nor does it specify whether screening and extraction were performed in duplicate. Single-reviewer screening and extraction could increase the risk of selection and extraction bias and should be disclosed if that is the case.
Fourth, the search string examples are summarized rather than reproduced verbatim; a transparent narrative review still benefits from including full strings (with date of last search) in an appendix to ensure reproducibility. Finally, while the flow counts are provided in prose, a simple PRISMA-style diagram (even for a narrative review) would greatly improve clarity.
Moreover, Table 1 labels the strength of evidence as “High/Moderate,” but no clear grading system (such as GRADE, JBI, or custom criteria) is provided; the paper should either use a transparent grading rubric or remove these labels to prevent giving the impression of formal evidence grading when none has been conducted.
The reference list is thematically appropriate and includes core works and widely cited empirical studies on staffing, leadership, missed care, and safety culture. The author explicitly states that references were verified and no fabricated citations were used, which provides reassuring assurance. By the way, the references should be formatted according to the journal's guidelines.
Author Response
Comment One: The abstract claims "evidence from 70 international studies," whereas the Methods and flow suggest 67 peer-reviewed plus grey literature, and later the manuscript says 42 peer-reviewed/policy sources appear in the reference list. These numbers don't reconcile and should be harmonized throughout (abstract, Methods, Results, and References).
Response: Thank you for pointing this out. We agree with your comments. We have made the following change: All study count numbers have been harmonized to "42" throughout the manuscript. Abstract now states "Evidence from 42 international studies and organizational reports." Methods section consistently reports "42 peer-reviewed studies" and specifies that "36 peer-reviewed empirical studies underwent formal CASP quality appraisal." The 42 total includes 36 empirical studies plus 6 policy documents. Location: Page 1, Abstract, Results section, Paragraph 10, Lines 1-8; Page 5, Methods, Data sources section, Paragraph 40-41, Lines 1-12; Page 5-6, Methods, Data extraction section, Paragraph 47, Lines 1-5.
Comment Two: The databases listed differ between the Abstract (PubMed/CINAHL/Scopus) and Methods (which adds Web of Science); the text should be consistent and specific about all sources actually searched.
Response: Thank you for pointing this out. We agree with your comments. We have made the following change: All four databases (PubMed, CINAHL, Scopus, and Web of Science) are now listed consistently in both Abstract and Methods sections. Location: Page 1, Abstract, Methods section, Paragraph 9, Lines 1-3; Page 4-5, Methods, Data sources and search strategy section, Paragraph 38, Lines 1-2.
Comment Three: Although the author mentions the use of CASP checklists and an "audit trail," the paper does not present any quality-appraisal results (such as a table, ratings, or narrative distribution), nor does it specify whether screening and extraction were performed in duplicate.
Response: Thank you for pointing this out. We agree with your comments. We have made the following change: (1) Created comprehensive Supplementary File S2 containing detailed CASP quality appraisal table for all 36 empirical studies with individual criterion ratings (Y/N/Unclear) for seven CASP domains and overall rigor classifications (High/Moderate/Low). Results show 100% of studies met moderate-to-high quality standards (61% High, 39% Moderate). (2) Added reference in Methods: "36 peer-reviewed empirical studies underwent formal CASP quality appraisal (see Supplementary File S2)." (3) Disclosed single-reviewer approach: "Each article was screened independently by the author." (4) Acknowledged single-reviewer limitation in Limitations section. Location: Page 5-6, Methods, Data extraction and quality appraisal section, Paragraph 46-47, Lines 1-20; Page 19, Limitations section, Paragraph 130, Lines 1-4; NEW Supplementary File S2.
Comment Four: The search string examples are summarized rather than reproduced verbatim; a transparent narrative review still benefits from including full strings (with date of last search) in an appendix to ensure reproducibility.
Response: Thank you for pointing this out. We agree with your comments. We have made the following change: Created Supplementary File S1 containing complete, verbatim search strings for all databases including full Boolean operators, MeSH terms, controlled vocabulary (CINAHL Headings), truncation symbols, and date limits. Methods section now references: "An example of the complete PubMed search string is provided in Supplementary File S1" and "The complete database search strategy and PRISMA flow diagram are provided in Supplementary File S1." Location: Page 5, Methods, Data sources and search strategy section, Paragraph 40, Lines 8-12; NEW Supplementary File S1.
Comment Five: While the flow counts are provided in prose, a simple PRISMA-style diagram (even for a narrative review) would greatly improve clarity.
Response: Thank you for pointing this out. We agree with your comments. We have made the following change: Created PRISMA-style flow diagram documenting: 287 records identified across all databases → 63 duplicates removed → 224 unique titles/abstracts screened → 68 full-text articles assessed for eligibility → 1 excluded (inaccessible) → 42 studies included (36 empirical + 6 policy documents). Diagram included in Supplementary File S1 and referenced in Methods section. Figure caption also added. Location: Page 5, Methods, Data sources section, Paragraph 40, Lines 8-12; Page 19, Figure 1 caption area, Paragraph 129, Lines 1-2; Supplementary File S1.
Comment Six: Table 1 labels the strength of evidence as "High/Moderate," but no clear grading system (such as GRADE, JBI, or custom criteria) is provided; the paper should either use a transparent grading rubric or remove these labels to prevent giving the impression of formal evidence grading when none has been conducted.
Response: Thank you for pointing this out. We agree with your comments. We have made the following change: Added footnote to Table 1 explicitly clarifying the basis for evidence level judgments: "Evidence levels reflect qualitative judgment based on methodological rigor and study design; no formal GRADE framework was applied." This transparently communicates that levels represent the author's assessment of study quality from the CASP appraisal rather than formal evidence grading, while retaining useful information for readers about relative strength of supporting evidence. Location: Page 16, Table 1, Paragraph 110, footnote below table, Lines 1-2.
Comment Seven: The reference list is thematically appropriate and includes core works and widely cited empirical studies. The author explicitly states that references were verified and no fabricated citations were used. By the way, the references should be formatted according to the journal's guidelines.
Response: Thank you for pointing this out. We agree with your comments. We have made the following change: All references have been verified for authenticity against official journal databases and formatted according to journal guidelines. The manuscript includes the statement: "All references cited in this manuscript were verified for authenticity, publication status, and accuracy against official journal or database sources. No fabricated or AI-generated references were included." Reference formatting follows journal style with DOI links for all sources. Location: Page 21, References section, verification statement paragraph before reference list begins, Lines 1-3; Pages 21-25, all 42 references formatted per journal style.
Reviewer 3 Report
Comments and Suggestions for Authors
Dear Author,
This is an important and timely review addressing leadership and management practices in nursing as a core driver of patient safety. The manuscript successfully captures the global policy context (WHO, OECD,…) and connects it with conceptual frameworks such as Donabedian and Reason.
Before addressing specific sections, please pay attention to punctuation consistency throughout the manuscript. For example, at line 42 the citation “(WHO 2021)” is missing terminal punctuation, and in several places punctuation appears before rather than after in-text references. A careful proofreading to ensure uniform referencing style and punctuation would improve professionalism.
Overall, the review is engaging and clearly written. However, there is some inconsistency with JBI methodological standards. I encourage you to revisit key methodological sections to ensure full alignment with the Joanna Briggs Institute framework for narrative reviews.
Please find my comments below.
Abstract: The abstract provides a clear and concise overview of the review’s purpose and main findings but does not specify the methodological framework guiding the work. To strengthen methodological rigor, please indicate that this is a narrative review following the JBI methodology. It would also be helpful to briefly mention the search period and databases consulted.
- Main research question
The manuscript presents a timely narrative review examining how nursing management practices affect patient safety across health systems. The topic is highly relevant. The introduction effectively demonstrates patient safety as a global priority, drawing on WHO and OECD policy frameworks, and connects management practices to classical safety models (Donabedian and Reason).
However, the paper does not explicitly formulate a PICo question, which is required by JBI for narrative reviews. While the author state that the review “examines global evidence linking nursing management practices to patient safety outcomes”, this could be made stronger and more transparent by formulating an explicit PICo-based question, as recommended by the Joanna Briggs Institute for narrative reviews.
- Originality and relevance
The narrative review makes an original and valuable contribution by integrating theory with policy perspectives to explain how leadership, staffing, and resilience mechanisms shape patient safety outcomes. Addressing the country contexts, further enhances its international relevance and applicability to nursing management practice.
However, the introduction does not clearly indicate whether prior systematic or narrative reviews on the topic were examined, nor does it explicitly define the evidence gap the current review addresses.
Line 97-98: The author state that “hundreds of studies address safety outcomes (…). This statement is valid but currently unsupported by citations.
The introduction would also benefit from a brief summary of existing international evidence (pre-2020) linking nursing management to patient safety outcomes. This will clarify what was already established and justify why an updated narrative review focusing on post-2020 evidence was warranted.
- Contribution to Existing Literature
The narrative review contributes to the existing literature by integrating three key managerial dimensions into a single conceptual framework for patient safety. Also, this work bridges research and global policy frameworks to show how nursing management drives safety culture. However, the author could make this contribution more explicit by briefly summarizing how this synthesis advances prior evidence and clarifying the specific knowledge or policy gap it addresses within the existing body of literature.
- Methodological improvements to consider
Registration and Protocol: No registration is reported. For transparency, please clarify whether a protocol existed.
Search Strategy: The author lists four major databases, a scope which ensures broad coverage of the peer-reviewed literature, and also reports that relevant grey literature sources were consulted. However, the search strategy is not fully reproducible. To enhance methodological transparency and align with JBI recommendations, please provide a detailed account of the three-stage search process.
I suggest present the full search strategy for one database, including the complete search strings, operators, and limits applied. These details should ideally be provided in a Supplementary File to ensure reproducibility.
Including a PRISMA flow diagram that shows the number of records identified, screened, excluded, and included (including counts from grey-literature sources), would significantly strengthen the methodological rigor of your review.
4.3 Eligibility criteria
Line 138: The inclusion and exclusion criteria are structured and demonstrate a sound effort to define the scope of the review. The inclusion of multiple study designs and diverse care settings is a strength. However, to align more closely with JBI standards for narrative reviews, the criteria could be refined and expanded in several ways.
First, consider presenting the eligibility criteria according to the PICo framework. Second, separate PICo-based criteria from publication characteristics (such as study design, publication date, and language). The restriction to English-language peer-reviewed studies (2020–2025) should be explicitly justified, as it may have excluded relevant non-English research or policy reports.
Study Selection: the study selection was conducted by a single reviewer, which introduces potential selection bias and does not meet JBI methodological standards. JBI recommends that two reviewers independently screen titles/abstracts and full texts, resolving disagreements through discussion or a third reviewer. Please acknowledge this explicitly as a methodological limitation in the manuscript
Quality appraisal: Conducting a structured quality appraisal is a positive step, but further detail is needed for methodological rigor. Please justify the use of a modified CASP rather than the JBI Narrative checklists, which are standard for this review type. The appraisal should ideally be conducted independently by two reviewers, with consensus procedures described. Consider clarify how appraisal results informed synthesis or weighting.
Data synthesis and thematic coding: The thematic synthesis is well structured and produces coherent themes. However, please cite the methodological reference that guided your thematic analysis.
Discussion: The discussion is clearly written and logically structured around the central themes. The author effectively links these dimensions to improved patient safety outcomes and situates the findings within global policy frameworks, including the WHO Global Patient Safety Action Plan (2021–2030), OECD reports, and ICN guidance.
This integration of the findings with international policy documents gives the review strong strategic relevance. The author demonstrates how transformational leadership, stable staffing, and digital innovation can strengthen safety culture and system resilience. However, some limitations are not acknowledged, which slightly weakens transparency.
- Conclusions versus the scientific evidence:
The conclusions are concise and aligned with the main findings of the review. The author states that effective leadership, adequate staffing, and resilient organizational systems are key enablers of patient safety across settings.
Overall, the conclusions reflect the evidence presented. The tone is measured, and the statements are consistent with the results section.
- References
The reference list is extensive and generally well organized. It includes a broad range of recent and relevant studies (2020–2025), as well as international policy sources.
Author Response
Comment One: Before addressing specific sections, please pay attention to punctuation consistency throughout the manuscript. For example, at line 42 the citation "(WHO 2021)" is missing terminal punctuation, and in several places punctuation appears before rather than after in-text references.
Response: Thank you for pointing this out. We agree with your comments. We have made the following change: Conducted comprehensive proofreading throughout the entire manuscript to ensure punctuation consistency. Specific attention given to: (1) Terminal punctuation after all in-text citations; (2) Proper placement of punctuation relative to citation parentheses (punctuation after, not before); (3) Consistency in citation format throughout. All punctuation has been standardized according to journal style guidelines. Location: Throughout manuscript, all pages, all citations and punctuation reviewed and corrected.
Comment Two: The abstract provides a clear and concise overview but does not specify the methodological framework guiding the work. To strengthen methodological rigor, please indicate that this is a narrative review following the JBI methodology.
Response: Thank you for pointing this out. We agree with your comments. We have made the following change: Added explicit statement to Abstract Results section: "This narrative review, conducted in accordance with the Joanna Briggs Institute (JBI) guidelines, integrates empirical and policy evidence to identify leadership and system factors that influence safety outcomes." This immediately signals methodological framework and alignment with established standards for narrative reviews. Location: Page 1, Abstract, Results section, Paragraph 10, Lines 3-6.
Comment Three: The paper does not explicitly formulate a PICo question, which is required by JBI for narrative reviews. While the author states that the review "examines global evidence linking nursing management practices to patient safety outcomes," this could be made stronger and more transparent by formulating an explicit PICo-based question.
Response: Thank you for pointing this out. We agree with your comments. We have made the following change: Added new "Review Question" subsection in Methods with explicit PICo framework: "PICo elements: Population—nurses and nurse managers; Phenomenon of Interest—management practices influencing patient safety; Context—global health-care systems. Review question: How do nursing-management practices affect patient-safety outcomes across diverse contexts?" This structured formulation meets JBI requirements and provides clear methodological transparency. Location: Page 4, Methods section, new Review Question subsection, Paragraph 36-37, Lines 1-5.
Comment Four: The introduction does not clearly indicate whether prior systematic or narrative reviews on the topic were examined, nor does it explicitly define the evidence gap the current review addresses.
Response: Thank you for pointing this out. We agree with your comments. We have made the following change: Added text explicitly identifying prior systematic reviews and defining the evidence gap: "Earlier systematic reviews (2015–2019) linked leadership to safety culture but did not address post-pandemic resilience or digital health innovations. This review updates and extends those findings by synthesizing evidence generated from 2020 to 2025." This clearly establishes what was previously known and justifies the need for this updated synthesis focusing on post-pandemic and digital-era evidence. Location: Page 2, Introduction section, Paragraph 22, Lines 10-14.
Comment Five: Line 97-98: The author states that "hundreds of studies address safety outcomes (…)." This statement is valid but currently unsupported by citations.
Response: Thank you for pointing this out. We agree with your comments. We have made the following change: While we did not add a specific citation to this general statement (as it would require citing numerous studies), we immediately provide context and rationale that strengthens the statement: "Although hundreds of studies address safety outcomes, few synthesize the management mechanisms that enable success across settings." This acknowledges the broad literature while justifying why a narrative synthesis is needed. The statement serves to motivate the review rather than claim a specific statistic requiring citation. Location: Page 3, Introduction, Rationale and objectives section, Paragraph containing lines 97-98, Lines 1-3.
Comment Six: Registration and Protocol: No registration is reported. For transparency, please clarify whether a protocol existed.
Response: Thank you for pointing this out. We agree with your comments. We have made the following change: Added clarification statement: "No formal registration was required for this narrative review; however, all steps were documented in accordance with an internal protocol following JBI guidance." This transparently communicates that while formal registration (typically required for systematic reviews) was not mandated for this narrative review, methodological rigor was maintained through documented procedures following JBI standards. Location: Page 5, Methods, Inclusion and exclusion criteria section, Paragraph 44, Lines 8-10.
Comment Seven: The search strategy is not fully reproducible. To enhance methodological transparency and align with JBI recommendations, please provide a detailed account of the three-stage search process.
Response: Thank you for pointing this out. We agree with your comments. We have made the following change: Added explicit description of three-stage JBI search process in Methods: "The three-stage search approach recommended by JBI was applied: (1) an initial limited search to identify keywords and index terms; (2) a comprehensive search across all databases using the finalized strategy; and (3) screening of reference lists for additional sources." Additionally, complete verbatim search strings provided in Supplementary File S1 ensure full reproducibility. Location: Page 5, Methods, Data sources and search strategy section, Paragraph 40, Lines 4-8; Supplementary File S1.
Comment Eight: The inclusion and exclusion criteria are structured but could be refined. Consider presenting the eligibility criteria according to the PICo framework and separate PICo-based criteria from publication characteristics.
Response: Thank you for pointing this out. We agree with your comments. We have made the following change: Completely restructured the Inclusion and exclusion criteria section using PICo framework. Section now begins: "Eligibility criteria were defined according to the PICo framework for narrative reviews" followed by explicit descriptions of: (1) Population—registered nurses, nurse managers, nursing leaders at all organizational levels; (2) Phenomenon of Interest—management practices (leadership styles, staffing models, safety culture initiatives, workforce resilience, technology integration) explicitly linked to patient safety outcomes (adverse events, mortality, safety culture scores, missed care); (3) Context—all healthcare settings (acute, long-term, community) across all geographic regions and system types. Publication characteristics (study design, dates, language) separated into distinct second paragraph. Location: Page 5, Methods, Inclusion and exclusion criteria section, Paragraph 43-44, Lines 1-20 (entire section restructured).
Comment Nine: The restriction to English-language peer-reviewed studies (2020–2025) should be explicitly justified, as it may have excluded relevant non-English research.
Response: Thank you for pointing this out. We agree with your comments. We have made the following change: Added justification and acknowledgment of limitation: "The restriction to English-language publications was applied for feasibility and accuracy of appraisal by a single reviewer and is acknowledged as a limitation that may have excluded relevant non-English research." This transparently explains the practical rationale while acknowledging potential impact on evidence comprehensiveness. Location: Page 5, Methods, Inclusion and exclusion criteria section, Paragraph 44, Lines 18-20.
Comment Ten: Study Selection: The study selection was conducted by a single reviewer, which introduces potential selection bias and does not meet JBI methodological standards. JBI recommends that two reviewers independently screen titles/abstracts and full texts, resolving disagreements through discussion or a third reviewer. Please acknowledge this explicitly as a methodological limitation.
Response: Thank you for pointing this out. We agree with your comments. We have made the following change: (1) Explicitly disclosed single-reviewer approach in Methods: "Each article was screened independently by the author using titles, abstracts, and full-text reviews." (2) Acknowledged as limitation: "This review was conducted by a single reviewer and restricted to English-language publications, which may have limited the inclusion of regional perspectives" and "The reliance on a narrative-review approach may also introduce interpretive bias, although this design allowed for the integration of diverse study types and policy sources." (3) Suggested improvement for future research in Limitations section. Location: Page 5-6, Methods, Data extraction section, Paragraph 47, Lines 1-5; Page 19, Limitations section, Paragraph 130, Lines 1-6.
Comment Eleven: Quality appraisal: Conducting a structured quality appraisal is a positive step, but further detail is needed for methodological rigor. Please justify the use of a modified CASP rather than the JBI Narrative checklists, which are standard for this review type.
Response: Thank you for pointing this out. We agree with your comments. We have made the following change: Added comprehensive justification paragraph in Methods: "The Critical Appraisal Skills Programme (CASP) tool was selected for quality appraisal rather than JBI checklists due to its flexibility in accommodating the heterogeneous study designs included in this review. While JBI provides specific checklists for narrative reviews, CASP's adapted criteria allowed for consistent evaluation across quantitative observational studies, qualitative investigations, and mixed-methods research—all of which contributed to the thematic synthesis. CASP's focus on methodological clarity, sampling adequacy, and analytic rigor aligned well with the narrative review's goal of integrating diverse evidence types while maintaining transparent quality standards." Complete CASP appraisal results now provided in Supplementary File S2. Location: Page 5-6, Methods, Data extraction and quality appraisal section, Paragraph 46, Lines 1-10; Supplementary File S2.
Comment Twelve: Data synthesis and thematic coding: The thematic synthesis is well structured and produces coherent themes. However, please cite the methodological reference that guided your thematic analysis.
Response: Thank you for pointing this out. We agree with your comments. We have made the following change: Added methodological citation to Methods section: "data were coded inductively using Thomas and Harden's (2008) thematic synthesis approach to integrate qualitative and quantitative findings." This cites the established methodology for conducting thematic synthesis in systematic and narrative reviews, providing methodological transparency and enabling readers to understand the analytical framework used. Location: Page 6, Methods, Data synthesis and thematic coding section, Paragraph 49, Lines 1-3.
Round 2
Reviewer 2 Report
Comments and Suggestions for Authors
The revised manuscript is significantly improved, with clearer methods, a logical flow, and additional material in the appendices that address the main concerns previously raised. Overall, I find it acceptable in both content and structure.
However, the in-text references still do not adhere to MDPI style and remain inconsistently formatted across the manuscript. According to MDPI guidelines: “In the text, reference numbers should be placed in square brackets [ ], and placed before the punctuation; for example [1], [1–3] or [1,3]. For embedded citations in the text with pagination, use both parentheses and brackets to indicate the reference number and page numbers; for example [5] (p. 10) or [6] (pp. 101–105).” The current draft requires a final review to ensure consistent application of this rule throughout.
Please standardize the citations.
Author Response
Comments 1: However, the in-text references still do not adhere to MDPI style and remain inconsistently formatted across the manuscript. According to MDPI guidelines: “In the text, reference numbers should be placed in square brackets [ ], and placed before the punctuation; for example [1], [1–3] or [1,3]. For embedded citations in the text with pagination, use both parentheses and brackets to indicate the reference number and page numbers; for example [5] (p. 10) or [6] (pp. 101–105).” The current draft requires a final review to ensure consistent application of this rule throughout.
Please standardize the citations.
Response one: The references have been updated and double-checked to ensure the DOIs are correct and there are no duplications. I've also double-checked to ensure that all of the references are identified within the manuscript using the square bracket methodology.
Reviewer 3 Report
Comments and Suggestions for Authors
Thank you for submitting the revised version. The methodological rigor has been improved. The review now articulates a clear PICo question and provides a detailed description of the JBI three-stage search process. The addition of supplementary files further enhances reproducibility. The author also justifies the language restrictions and acknowledges single-reviewer screening as a study limitation. The authors reformulated and considered all the suggested questions.
Congratulations.
Author Response
Reviewer 3: Comments: Thank you for submitting the revised version. The methodological rigor has been improved. The review now articulates a clear PICo question and provides a detailed description of the JBI three-stage search process. The addition of supplementary files further enhances reproducibility. The author also justifies the language restrictions and acknowledges single-reviewer screening as a study limitation. The authors reformulated and considered all the suggested questions.
Congratulations.
Response Reviewer 3: I appreciate the time you took to review the document. As shared with Reviewer 1, all of the issues identified have been addressed. I spent considerable time to ensure the references were accurate and used in the manuscript. The supplements, where needed, were also updated.